# Differences between Systems Using Optical and Capacitive Sensors in Treadmill-Based Spatiotemporal Analysis of Level and Sloping Gait

**DOI:** 10.3390/s22072790

**Published:** 2022-04-05

**Authors:** Dimitris Mandalidis, Ioannis Kafetzakis

**Affiliations:** Sports Physical Therapy Laboratory, Department of Physical Education and Sports Science, School of Physical Education and Sports Science, National and Kapodistrian University of Athens, 17237 Athens, Greece; kafetzos13@icloud.com

**Keywords:** inter-system agreement, treadmill walking, gait analysis, inclined surfaces

## Abstract

Modern technology has enabled researchers to analyze gait with great accuracy and in various conditions based on the needs of the trainees. The purpose of the study was to investigate the agreement between systems equipped with optical and capacitive sensors in the analysis of treadmill-based level and sloping gait. The spatiotemporal parameters of gait were measured in 30 healthy college-level students during barefoot walking on 0% (level), −10% and −20% (downhill) and +10% and +20% (uphill) slopes at hiking-related speeds using an optoelectric cell system and an instrumented treadmill. Inter-system agreement was assessed using the Intraclass Correlation Coefficients (ICCs) and the 95% limits of agreement. Our findings revealed excellent ICCs for the temporal and between moderate to excellent ICCs for the spatial parameters of gait. Walking downhill and on a 10% slope demonstrated better inter-system agreement compared to walking uphill and on a 20% slope. Inter-system agreement regarding the duration of gait phases was increased by increasing the number of LEDs used by the optoelectric cell system to detect the contact event. The present study suggests that systems equipped with optical and capacitive sensors can be used interchangeably in the treadmill-based spatiotemporal analysis of level and sloping gait.

## 1. Introduction

Quantifying data of the gait cycle has been a valuable asset to many clinical therapists in making decisions about the effects of injury and disease on patients’ functional ability, as well as in monitoring the therapeutic interventions [1]. These data are mainly collected by systems equipped with sensors that provide information regarding the spatiotemporal and/or dynamic gait parameters both during over-ground, and treadmill walking and running. Among the systems that have been frequently used in gait analysis over the last decade, for clinical as well as for research purposes, are those consisting of optical and capacitive sensors. One such system is the optoelectric cell system (OCS), known as Optogait (Microgate S.r.I, Bolsano, Italy) which consists of optical sensors embedded into bars. The sensors can detect any interruption in the light signal due to the presence of feet within the recording area, as a person walks or runs between pairs of bars that have successively connected, parallel to each other, on the ground. Apart from the fact that the system calculates reliably the spatiotemporal parameters during over-ground walking as it is naturally performed by an individual [2,3], it is portable, and therefore, it can be used on all flat surfaces and is less costly. The great length of the walking path and the size of the working area required for over-ground walking, as well as the inability to reproduce the gait conditions are some of the limitations of the system. However, many of the above limitations are eventually canceled out by placing a pair of bars on the sides of a standard treadmill [4,5,6]. In this case, the gait analysis can be performed under the standardized conditions that a treadmill can offer (e.g., predetermined and controlled speed, slope, number of steps, etc.) as opposed to over-ground walking.

The spatiotemporal gait parameters as well as the distribution of pressures exerted during treadmill walking and running have also been measured with instrumented treadmills (ITR). These are treadmills equipped with a measuring matrix consisting of capacitive pressure sensors embedded beneath the running belt. A capacitive sensor consists of two plates made of a conducting material separated by a non-conducting or insulating layer termed a “dielectric”. When a force is applied to the electrically charged sensor, a change in voltage is recorded as the two plates are compressed, reducing the distance between them and increasing the capacitance [7]. Apart from being a valid means for gait performance, at least from a kinetic and kinematic perspective [8], ITRs allow continuous measurements of the spatiotemporal and dynamic gait parameters under predetermined and standardized conditions, thus enabling comparisons between different populations, and provide reliable parameters both for level [9,10,11] and sloped walking [12]. Instrumented treadmills require less space, as all treadmills do, but they are far more expensive compared to systems that analyze over-ground walking.

One of the features of the treadmill, which can eventually be utilized in conjunction with the Optogait OCS, is the adjustment of its surface in different slopes. In this context, the Optogait OCS has been used in a few studies, mainly during uphill walking and running by healthy individuals [13,14,15], though the validity of the system in these conditions remains uncertain. So far, the validity of the Optogait OCS has been established for both over-ground [16,17,18] and treadmill walking [19] and running [20], but only on level surfaces (0% slope). Even though the validity of the system was found acceptable for clinical and research purposes under the specific experimental conditions, various reasons prevent researchers from relying on the available information to analyze treadmill-based gait on sloped surfaces. Firstly, because several spatial and temporal gait parameters recorded during over-ground walking, either on a level surface [21] or on a ramp [22], are not representative of those obtained from an instrumented treadmill when walking on a flat or sloping surface. Secondly, because the Optogait OCS has shown excellent concurrent validity compared to an instrumented treadmill for level walking at self-selected speeds, but only for speed, step and stride length, step and stride time, and cadence. The inter-system agreement for the duration of single and double limb support, and the stance and swing phases was low suggesting that the two devices cannot be used interchangeably [19]. Thirdly, because the validity of gait measurements can vary when a person is forced to walk at a certain speed on inclined surfaces where it may not be too accustomed.

The agreement between gait analysis systems allows researchers to use them interchangeably depending on their availability. In the absence of an ITR, the use of the less expensive OCS, such as Optogait, in conjunction with a treadmill can provide clinicians and researchers the ability to analyze gait in various and/or more demanding conditions, maintaining the advantages that the treadmill-based gait offer. Therefore, the purpose of this study was to determine the agreement between the Optogait OCS and the INT in the analysis of level and sloping gait.

## 2. Materials and Methods

### 2.1. Study Sample

The study sample consisted of 30 healthy, physically active collegiate students (13 males and 17 females, mean ± SD age of 25.1 ± 3.8 y, height: 1.7 ± 0.1 m, body weight: 66.2 ± 12.0 kg and BMI: 22.8 ± 2.2 kg m^−1^). Volunteers with pain, inability to fully bear their body weight or limping while walking, those with a history of neurological, visual, vestibular, or balance disorders, and those who expressed fatigue or discomfort while performing the study protocol, were excluded from the study. Each of the selected volunteers was informed of the purpose of the study and signed a written consent before the commencement of the testing procedure. 

### 2.2. Instrumentation

A treadmill (Pluto^®^ Med, h/p/cosmos^®^ Sports & Medical GmbH, Nussdorf–Traunstein, Germany) with a 150 cm (L) × 50 cm (W) running surface and a capacitance-based pressure platform (FDM- THPL-M-3i, Zebris Medical GmbH, Isny, Germany) embedded beneath the running belt was used in this study (Figure 1). The slope and speed settings of the treadmill allowed level and uphill walking on slopes ranging between 0.1–20.0% using speeds up to 18.0 km h^−1^. The reverse rotation feature of the treadmill’s belt allowed downhill walking on slopes ranging between 0.1%–20.0% at speeds up to 5.0 km h^−1^. The sensor area of the pressure platform (L: 108.4 × W: 47.4 cm) consisted of 7168 sensors which collected data at a sampling rate of 240 Hz. The sensor threshold was set at 1 N cm^−2^.

The Optogait OCS used in this study consisted of an emission and a receiving bar 100 cm (L) × 8 cm (W) in size that were placed on the sides of a treadmill’s frame with the drums of the bars facing towards the front side of the treadmill (Figure 1). Based on this placement of the bars, the direction parameter for level and uphill walking was set to “Interface side”, and when the rotation of the treadmill’s belt was reversed for downhill walking, the direction parameter was set to “Opposite side”. The two bars communicated with each other at an infrared frequency via 96 LEDs that are placed 1 cm apart and 3 mm above the floor level. The spatiotemporal gait parameters were calculated by detecting the communication interruptions between the bars caused by the participant’s movements. The data were collected at a sampling frequency of 1000 Hz and transmitted to a personal computer where they were stored for later analysis with dedicated Optogait software (version 1.6.4.0, Microgate S.r.I, Bolsano, Italy). Spatiotemporal gait parameters in all conditions were calculated by setting the GaitR In and GaitR Out filter parameters to zero and they were re-calculated after re-setting both parameters at 0, 1, 2, 3 and 4. The specific setting indicated that the minimum number of LEDs that were interrupted for triggering the contact event were 1, 2, 3, 4 and 5, respectively (Figure 2). The starting foot that is the first foot that interrupted the communication between the bars, was detected after the gait session was completed, as each participant was required to have reached the pre-determined gait speed before the start of the test. This was accomplished by video recording each gait condition using a webcam (Logitech c920 pro HD stream), which was synchronized with the Optogait system. The webcam was placed on the side of the treadmill and connected to the USB port of a personal computer. Recognition of the starting foot was achieved by displaying the first frame of the gait, immediately after the initiation of the test, via the “video preview popup” feature of the software. 

### 2.3. Testing Procedure

Each participant required to walk barefoot on the treadmill with its surface placed at 0% (level), −10% and −20% (downhill) and +10% and +20% (uphill) slopes. Gait on 0% and −10% slopes was performed with 5.0 km h^−1^, on +10% and −20% slopes with 3.5 km h^−1^ and on a +20% slope with 2.5 km h^−1^. The slopes were selected based on the slopes of the surfaces commonly encountered in urban residential areas or on trails proposed for recreation [23,24]. Speeds were selected based on the average gait speed for males and females 20–39 years of age [25] and the Tobler’s hiking exponential function determining the hiking speed, taking into account the slope angle [26,27].

The gait protocol included a 4-min walk for familiarization in the predetermined slope/speed and another 4-min walk for the actual test, the data of which were used in the statistical analysis. The familiarization period was selected based on previous studies showing that this period of time is sufficient for treadmill acclimatization to reach a stable performance for most of the spatiotemporal parameters of gait [28]. It also aimed to prevent fatigue and be time-efficient, given the number of conditions under which gait was analyzed. A 2-min break was given between the familiarization period and the actual test and a 4-min rest was kept between walking in different slopes/speeds (Figure 2). 

Nevertheless, to avoid fatigue, participants performed all tests included in the study protocol in a random order. This was accomplished by instructing each participant to choose a number between 1 and 120 with each of the numbers representing a group of gait tests arranged in a different and random order. The 120 different combinations of the five walking conditions (at 0%, −10%, −20%, +10%, +20% slopes) were randomly arranged in a sequence created with a web application (https://www.random.org, accessed on 16 January 2022). Furthermore, fatigue during the testing procedure was monitored by recording the heart rate (HR) and the perceived excursion before the start of each gait session. In the event that a participant’s HR exceeded 60% of the maximum HR or 17 points on the Borg scale, the test was interrupted and the participant was dismissed. These two parameters have been associated with loss of postural control [29] and severe excursion [30], respectively, and could affect participants’ ability to walk normally, especially under demanding uphill and downhill conditions. Participants were instructed to abstain from strenuous activities before reporting to the laboratory for testing and to wear lightweight and comfortable clothing and having their gaze look straightforward.

### 2.4. Data Analysis

Data were analyzed based on the mean value obtained from the steps performed during each 4-min walk for the spatial (e.g., step and stride length) and the temporal parameters (e.g., step and stride time, cadence, duration of gait phases) recorded by the Optogait OCS and the ITR (Table 1). All gait parameters corresponded to the average of the right and left foot values (where possible). 

### 2.5. Statistical Analysis

The agreement between the two gait analysis systems for the spatiotemporal gait parameters was assessed using two-way fixed effects Intraclass Correlation Coefficients (ICC (3,k)). The ICC model (3,k) was used in the context that the two systems used for gait analysis, i.e., the Optogait OCS and the ITR developed by Zebris, were the only ones under investigation with k, the mean value for each gait parameter calculated based on the number of steps performed in each walking condition. ICCs were interpreted as excellent (>0.90), good (0.75–0.90), or poor to moderate (<0.75) [31]. The absolute agreement was expressed with the systematic bias (SB) and the 95% limits of agreement (95%LOA), that is, the mean difference and the standard deviations of the mean difference for the spatiotemporal gait parameters measured with the OCS and ITR [32,33]. The differences (i.e., SBs) between the measurements recorded with the OCS and the ITR were assessed using a paired t-test [32,33]. The differences between the LEDs used to trigger the contact event regarding the SBs obtained for all spatiotemporal gait parameters were assessed with one-way repeated measures ANOVA. Sphericity was determined based on the Mauchly’s Test, and the Greenhouse–Geisser correction was used when sphericity was significant. Significant main effects were followed by pairwise comparisons after controlling for type I errors using a Bonferroni adjustment. The level of statistical significance was set to *p* < 0.05. Statistical analyses were conducted in SPSS, version 26.0 (IBMCorp, Armonk, NY, USA).

## 3. Results

The results of the present study were analyzed based on data obtained from 27 of the 30 participants. One participant discontinued the test due to the formation of blisters in the sole of the foot while two others were excluded from the study due to a lack of familiarity with walking on the treadmill, especially at the 20% slope. None of the participants demonstrated signs of fatigue during the testing protocol. The perceived exertion reported before the start of each 4-min walk, based on Borg’s 15-point scale, ranged between 8.0 and 10.7 while the mean HR did not exceed 50% of HRmax.

### 3.1. Agreement between OCS and ITR for the Spatiotemporal Parameters of Gait at 0% Slope

The ICCs were excellent (>0.998) for the spatial parameters and ranged from good (0.865) to excellent (0.928) for the temporal parameters of gait at 0% slope. Significant systematic bias was found only for the duration of the gait phases when 1, 2, 3, and 4 LEDs were used for gait analysis with the OCS (see Table 2 for individual comparisons). Repeated measures ANOVA between the LEDs that were used for detection of the contact event revealed significant differences for the systematic bias associated with the step and stride length (*p* < 0.05), the cadence (*p* < 0.001), and the duration of the gait phases (*p* < 0.001) (see Table 2 for pairwise comparisons). In general, systematic bias and 95% LoA for step and stride length were increased and, for cadence and duration of gait phases, decreased (see Bland and Altman plot in Figure 3) as the number of LEDs used for gait analysis with OCS was increased. The systematic biases and 95% LoA for step and stride time were not affected by the number of LEDs used for gait analysis.

### 3.2. Agreement between OCS and ITR for the Spatiotemporal Parameters of Gait at −10% and −20% Slopes

The ICCs were excellent for all the spatial and temporal parameters (>0.90) of gait at a −10% slope. Systematic biases were significant only for stride length and the duration of gait phases regardless of the LEDs used for gait analysis with the OCS (see Table 3 for individual comparisons). Repeated measures ANOVA revealed significant differences between the LEDs used for gait analysis regarding the systematic bias of the step and stride length (*p* < 0.01), the cadence (*p* < 0.001), and the duration of gait phases (*p* < 0.001) (see Table 3 for pairwise comparisons). In general, systematic bias and 95% LoA for step and stride length were increased and, for cadence and duration of gait phases, decreased (see Bland and Altman plot in Figure 4) as the number of LEDs used for gait analysis with OCS was increased. The systematic biases and 95% LoA for step and stride time were not affected by the number of LEDs used for gait analysis.

The ICCs were excellent for the spatial parameters (>0.997) and ranged between good (0.864) to excellent (0.999) for the temporal parameters of gait at −20% slope. The systematic errors were significant only for the duration of gait phases regardless the number of LEDs use for gait analysis (see Table 4 for individual comparisons). Repeated measures ANOVA revealed significant differences between the LEDs used for detection of the contact event regarding the systematic bias of the duration of the gait phases (*p* < 0.001) (see Table 4 for paiwise comparisons). In general, systematic bias and 95% LoA for cadence and duration of gait phases decreased (see Bland and Altman plot in Figure 5) as the number of LEDs used for gait analysis with OCS was increased. The systematic biases and 95% LoA for step and stride length and time were not affected by the number of LEDs used for gait analysis.

### 3.3. Agreement between OCS and ITR for the Spatiotemporal Parameters of Gait at +10% and +20% Slopes

The ICCs were excellent for the spatial parameters (0.997), and ranged between good (0.866) to excellent (0.999) for the temporal parameters of gait at +10% slope. Significant systematic biases were obtained for the step and stride length and the duration of gait phases regardless the LEDs used for gait analysis with the OCS (see Table 5 for individual comparisons). Repeated measures ANOVA revealed significant differences between the LEDs used for detection of the contact event regarding the systematic bias of the step and stride length, and the cadence (*p* < 0.05) as well as the duration of gait phases (*p* < 0.001) (see Table 5 for paiwise comparisons). In general, systematic bias and 95% LoA for step and stride length were increased and, for cadence and duration of gait phases, decreased (see Bland and Altman diagram in Figure 6) as the number of LEDs used for gait analysis with OCS was increased. The systematic biases and 95% LoA for step and stride time were not affected by the number of LEDs used for gait analysis.

The ICCs were excellent for the spatial and temporal parameters (0.915) of gait at +20% slope. Significant systematic biases were obtained for the step and stride length, the cadence and the duration of gait phases regardless the LEDs used for gait analysis with the OCS (see Table 6 for individual comparisons). Repeated measures ANOVA revealed significant differences between the LEDs used for detection of the contact event regarding the systematic bias of the step time (*p* < 0.01), the stride time (*p* < 0.05), the cadence (*p* < 0.05) and the duration of gait phases (*p* < 0.001) (see Table 6 for paiwise comparisons). In general, systematic bias and 95% LoA for step and stride length were increased and, for cadence and duration of gait phases, decreased (see Bland and Altman diagram in Figure 7) as the number of LEDs used for gait analysis with OCS was increased. The systematic biases and 95% LoA for step and stride time were not affected by the number of LEDs used for gait analysis.

## 4. Discussion

The results of the present study showed excellent, and only in few cases, good ICCs between the two systems regarding the spatiotemporal parameters of treadmill-based level and sloping gait. The absolute agreement expressed by the SB and the 95% LoA between the OCS and the ITR was also minimal, and in several cases negligible or zero, for the majority of the measured parameters. The SB was significant in many cases but it did not exceed 0.6 cm for the temporal parameters, 0.1 steps min^−1^ for the cadence during gait in all but in ±20% inclinations, where the SB for cadence reached 0.5 steps min^−1^, and 0.1 s for the step and stride time, as well as the duration for the gait cycle phases. The SB and, consequently, the 95% LoA of spatiotemporal gait parameters were also affected by the number of LEDs used for gait analysis with the OCS. Both step and stride lengths during level and sloping gait were significantly but slightly increased, and the temporal parameters, particularly the duration of gait cycle phases, were decreased as the number of LEDs used to identify the contact event by the OCS was increased.

The excellent ICC values and the minimum SB and 95% LoA reported for most of the spatiotemporal parameters in the present study could be justified by the arrangement of the systems and the settings that were taken under consideration for gait analysis. The height measurements between the upper flat surface of the sides of the treadmill frame (19.4 cm) and the surface of the treadmill running belt to the ground (19.6 cm) showed that the belt was 2 mm higher than the sides of the frame. This constructional feature of this particular type of treadmill eventually eliminated, to some extent, the inherent height difference that the optical sensors of the OCS demonstrate (3 mm above the walking surface). The potential effect of the OCS’s optical sensors location a few millimeters above the ground on the temporal parameters of gait has been pointed out by several authors, as this configuration may engage or postpone the interruption of the sensors a few milliseconds before the heel touches the ground or a few milliseconds after toe off, respectively. This is one of the factors that, according to the same authors, may contribute to the systematic bias that has been found for the temporal parameters of treadmill-based walking and running. In this context, Lee et al. [19] reported excellent ICC values for the spatial and temporal gait parameters, such as the step and stride time and cadence, but not for the duration of gait cycle phases during level walking at a self-selected speed (0.85 m·s^−1^ or 3.1 km·h^−1^) in both healthy individuals and stroke patients. Weart et al. [20], in a more recent study, compared the Optogait OCS with the Bertec ITR and reported good (0.83) to excellent (0.99) ICCs for step rate, step length, and contact time during running at a minimum pace of 2.7 m s^−1^ (9.7 km h^−1^). However, despite the high ICC, the contact time was overestimated by the Optogait OCS in 29 out of the 30 participants.

According to the manufacturers’ instructions, the potential negative effect of sensors’ location on gait analysis, particularly when the Optogait OCS is compared with other gait analysis systems, can be regulated by determining the minimum number of sensors that triggers a contact event. This can be achieved by setting the built-in GaitR In and GaitR Out filters to 0, 1, 2, 3, 4, or more, which means that a contact event will be considered valid only when 1, 2, 3, 4, or 5 LEDs, are activated. Although the number of sensors can be set by researchers for gait analysis with the Optogait OCS, it is not always reported in studies investigating the agreement between this and other gait analysis systems [19]. Others, however, comparing different sensor settings recommend the use of the 2-LED filter setting if gait parameters from the OptoGait OCS are to be compared to a three-dimensional motion capture system for over-ground walking at a self-selected speed [18]. The same number of sensors (2 LEDs) is required to achieve good agreement between the Optogait OCS, power plates and high-speed videography for the contact and flight times achieved in over-ground walking with speeds ranging between 11 and 15 km h^−1^ [34]. However, it has been proposed that for the evaluation of these temporal parameters on a treadmill, the GaitR In and GaitR Out of the OCS should be set to 0. Using various combinations of sensors for the analysis of gait with the OCS in the present study, it was found that the systematic biases created between the measurements with the two systems gradually decreased as the number of sensors that were set to identify the contact event, increased. Eventually, the minimum number of LEDs needed to be used for valid calculations of gait parameters with the Optogait OCS compared to the Zebris ITR was 5. Apparently, the agreement between the two systems on gait parameters increased when a significant part of the hind and front foot was in contact with the treadmill running belt, thus exerting a significant load on the platform. More information on the methods used for the calculation of the temporal parameters of gait will enhance our understanding on the agreement between the two systems that were implemented in the analysis of gait.

The systematic bias obtained for the temporal parameters of treadmill-based gait can also be exacerbated by the force threshold used to determine initial contact and toe off on the instrumented treadmill [20]. Most researchers did not report the initial pressure threshold for the device used in gait analysis [17,19] and, when a threshold of 50 N was used to detect initial contact [20], the contact time was underestimated compared to the OCS, demonstrating a negative systematic bias in favor of the ITR. The inter-system agreement found in the present study may be greater compared to that reported elsewhere because the threshold of initial contact was set to 1 N cm^−1^, a value that according to previous evidence may be more appropriate for gait analysis with low walking speeds [35].

A technical feature that may have contributed to the agreement of the systems is the sampling frequency used for gait analysis with the Zebris ITR. It is generally recommended that a sufficient sampling rate of the sensors to accurately measure the plantar pressure, when performing most daily walking and running activities, is 200 Hz [7]. The sampling frequency used in the present study was 240 Hz which was significantly higher compared to the sampling frequency (100 Hz) used in previous studies [19].

Our findings also revealed that the inter-system agreement for the temporal parameters of gait (duration of gait phases) was dependent on the direction (level, uphill or downhill) and the magnitude of the slope (10% or 20%). Based on the SB and the 95% LoA, the greatest agreement between gait analysis systems presented during level followed by uphill and downhill walking. Agreement was also better during walking on a 10% slope as opposed to walking on a 20% slope. Walking uphill or downhill is a demanding task that requires considerable effort [36,37], especially from people in poor physical condition. The fact that this activity is performed on a treadmill can further increase physical and psychological fatigue increasing the variability of the steps and, therefore, the temporal gait parameters calculations [38,39]. From a mechanical point of view, walking uphill or downhill can affect the movements of the lower limb joint to such an extent that it ultimately disrupts the normal gait phases, at least in relation to the way they are performed on a horizontal surface. Sarvestan et al. [40], for example, showed that the ankle joint may reach 16° of dorsiflexion angles when walking on a 10° (17.6%) uphill slope, while other researchers revealed that the range of motion of the ankle dorsiflexion decreases when walking downhill [38]. In these cases, the gait cycle may not start with a distinct heel contact, and the foot may appear flatter than usual and, therefore, more parallel to the bars of the Optogait OCS system. In people with limited dorsiflexion of the ankle, a condition that is very common in the general population, gait performance may be further impaired, increasing the likelihood of miscalculating temporal gait parameters [41,42]. Although the ROM of the ankle joint was not measured, it cannot be ruled out as one of the factors that influenced the temporal parameters of gait when walking uphill or downhill.

### Study Limitations

Considering the sample, the procedure and the technical characteristics of the sensors, the results of our study should be limited to the specific population, study protocol and sensors used by the gait analysis systems. The participants in the present study were active, but not necessarily accustomed to uphill or downhill walking. This was one of the reasons why some of them were excluded from the study. Nevertheless, their HR remained below 60% of HRmax in all walking conditions except with uphill walking at a 20% slope. In this condition, the average HR reached 67% of the participants’ HRmax, a factor that could affect postural control and, ultimately, the recording of gait phases [29]. Individuals of either gender and with various ages but with a better physical fitness level and/or more familiar with sloping gait, could maintain body stability and perform better on sloping gait, enabling systems to capture individual gait phases with more precision. Our findings are also limited to the number of steps performed per walking condition as we use different speeds for level, uphill and downhill walking. Eventually, the higher number of steps performed in our study during the 4-min walking may have contributed to the higher agreement between the systems compared to that reported in a previous study where gait was recorded for only 60 s at 0% slope [19]. Hence, it is possible that different number of steps may yield a different level of inter-system agreement. Finally, spatiotemporal gait measurements may be affected by changes in the sensitivity of the capacitive sensors embedded into the Zebris treadmill. Pathak and Ahn [43] showed that a 10-min break is not enough for the embedded sensors to recover their sensitivity after a 10-min walk at the preferred walking speed in measuring VGRF. In our study, a 2-min break was given between the two 4-min walking sessions (familiarization and trial) and a 4-min rest was allowed between walking in different slopes. This intermittent gait protocol may have helped maintain the necessary sensitivity of the sensors achieving ultimately high agreement between the gait analysis systems. However, more research is needed to identify the optimum break that should be given between walking sessions to maintain sensors’ sensitivity.

## 5. Conclusions

Gait analysis is part of an individual’s functional evaluation, the purpose of which is either to detect functional asymmetries or to monitor the progress of a rehabilitation program. Analyzing gait under more demanding conditions such as on sloping surfaces may enable clinicians to identify kinematic or electromyographic deficiencies in the joints and muscles of the lower extremity, respectively, that would otherwise remain unnoticed by walking on a flat surface [37,44]. Eligible for sloping gait analysis are also individuals who walk on sloping surfaces for everyday (e.g., residents in hilly urban or rural areas), professionals (e.g., rescue team members) or recreational purposes (e.g., hiking or mountaineering). The results of the present study showed that a lightweight, portable, adjustable and cost-effective system consisting of optical sensors can perform treadmill-based spatiotemporal gait analysis with approximately the same accuracy as an instrumented treadmill consisting of capacitive sensors. Clinicians and researchers should be aware of the adjustments that should be made on gait analysis systems (e.g., number of optical sensors to activate a contact event, sensitivity of capacitive sensors) and test protocols (e.g., adequate breaks between walking events) to achieve optimal gait analysis.

## Figures and Tables

**Figure 1 sensors-22-02790-f001:**
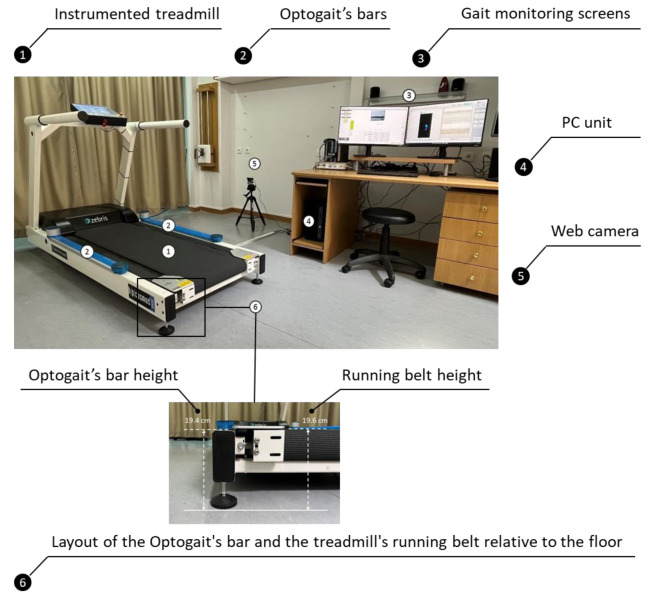
Experimental setup for the measurements of the spatiotemporal parameters of gait using the Optogait’s optoelectric cell system and the Zebris instrumented treadmill.

**Figure 2 sensors-22-02790-f002:**
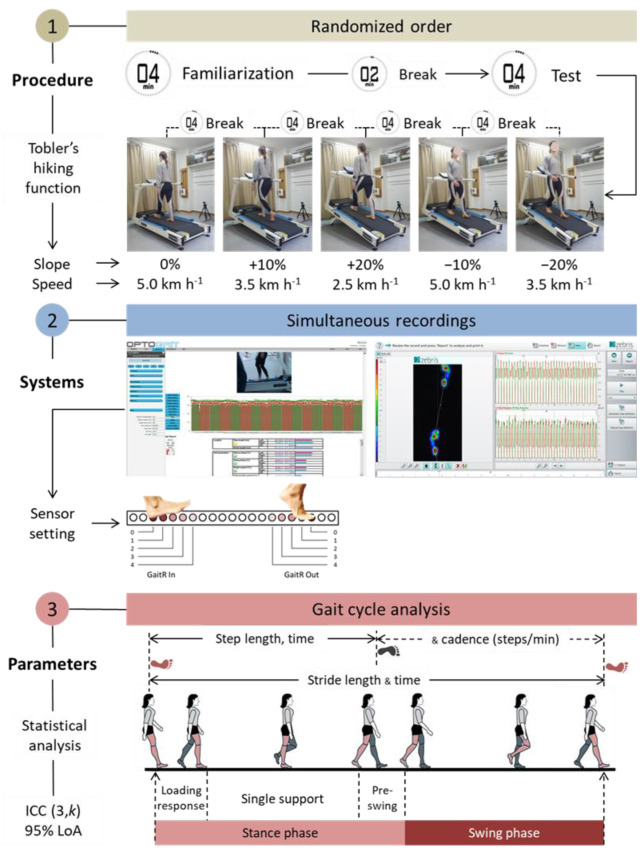
Schematics representing the steps of the experiment.

**Figure 3 sensors-22-02790-f003:**
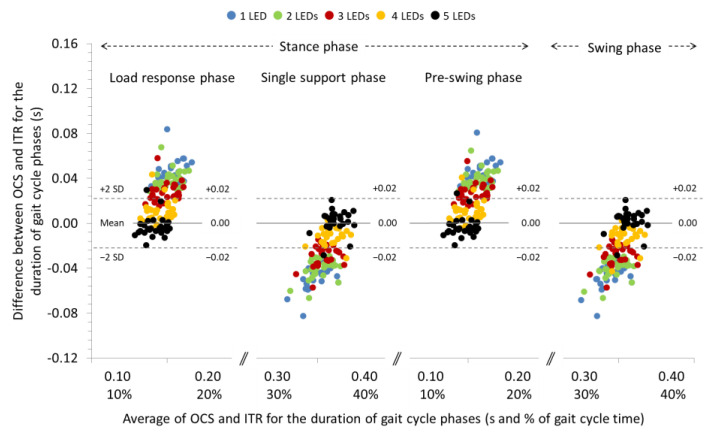
Bland and Altman plot depicting the systematic bias relative to the average duration of the gait cycle phases measured with the optoelectric cell system (OCS) using 1, 2, 3, 4, and 5 LEDs and the instrument treadmill (ITR), during walking at an inclination of 0%. The solid and dashed lines correspond to the systematic bias and 95% LoA, respectively, for the 5 LEDs setting.

**Figure 4 sensors-22-02790-f004:**
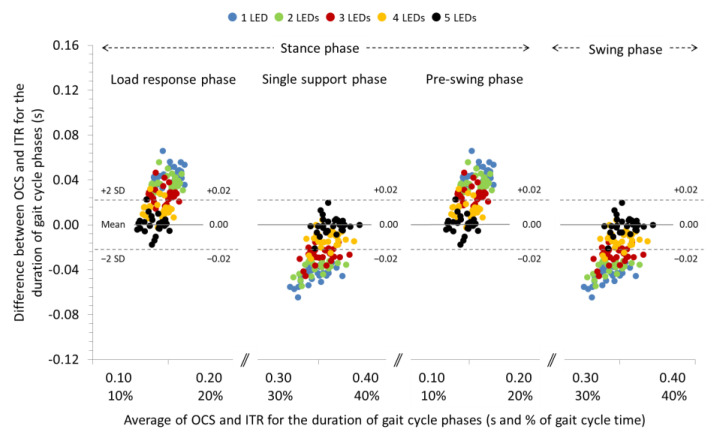
Bland and Altman plot depicting the systematic bias relative to the average duration of the gait cycle phases measured with the optoelectric cell system (OCS) using 1, 2, 3, 4, and 5 LEDs and the instrument treadmill (ITR), during walking at an inclination of −10%. The solid and dashed lines correspond to the systematic bias and 95% LoA, respectively, for the 5 LEDs setting.

**Figure 5 sensors-22-02790-f005:**
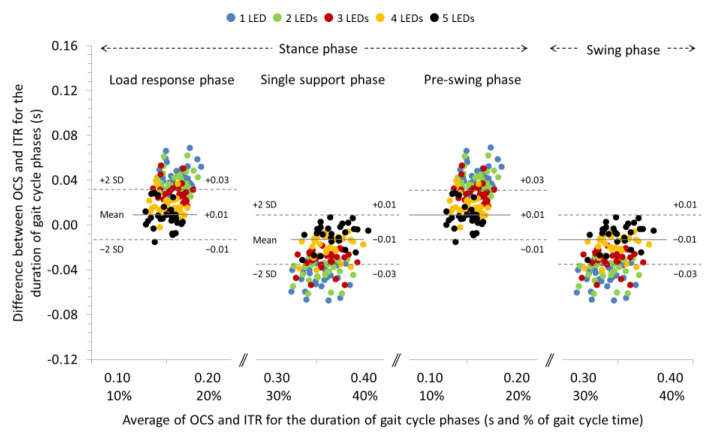
Bland and Altman plot depicting the systematic bias relative to the average duration of the gait cycle phases measured with the optoelectric cell system (OCS) using 1, 2, 3, 4, and 5 LEDs and the instrument treadmill (ITR), during walking at an inclination of −20%. The solid and dashed lines correspond to the systematic bias and 95% LoA, respectively, for the 5 LEDs setting.

**Figure 6 sensors-22-02790-f006:**
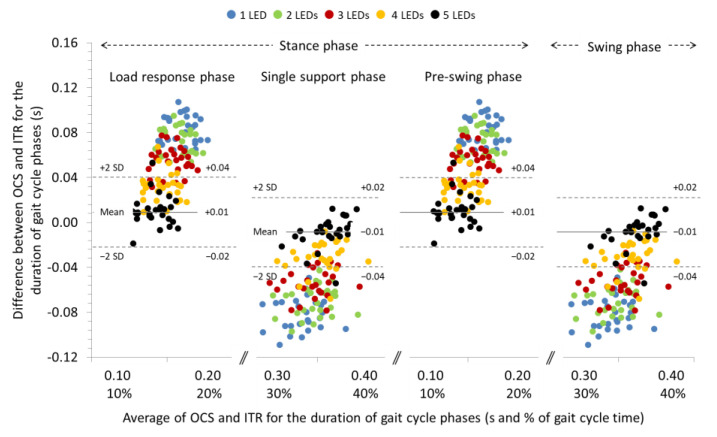
Bland and Altman plot depicting the systematic bias relative to the average duration of the gait cycle phases measured with the optoelectric cell system (OCS) using 1, 2, 3, 4, and 5 LEDs and the instrument treadmill (ITR), during walking at an inclination of +10%. The solid and dashed lines correspond to the systematic bias and 95% LoA, respectively, for the 5 LEDs setting.

**Figure 7 sensors-22-02790-f007:**
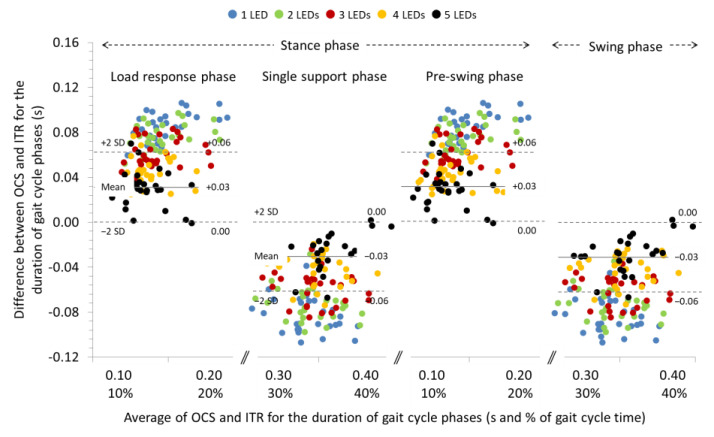
Bland and Altman plot depicting the systematic bias relative to the average duration of the gait cycle phases measured with the optoelectric cell system (OCS) using 1, 2, 3, 4, and 5 LEDs and the instrument treadmill (ITR), during walking at an inclination of +20%. The solid and dashed lines correspond to the systematic bias and 95% LoA, respectively, for the 5 LEDs setting.

**Table 1 sensors-22-02790-t001:** Definition of the spatial and temporal parameters measured by the photoelectric cell system and the instrumented treadmill.

Spatiotemporal Gait Parameters	Definition
Step length (cm)	The distance between the point of heel contact of one foot and the point of a successive heel contact of the contralateral foot
Stride length (cm)	The distance between the point of heel contact of one foot and the point of a successive heel contact of the same foot
Step time (s)	The time between the point of heel contact of one foot and the point of a successive heel contact of the contralateral foot
Stride time (s)	The time between the point of heel contact of one foot and the point of a successive heel contact of the same foot
Cadence (step/min)	The frequency of steps per unit time
Loading response phase (s)	The time between heel contact of one foot and just before toe off of the contralateral foot.
Single support phase (s)	The time during which the entire plantar aspect of the weight-bearing foot has contact with the ground.
Pre-swing phase (s)	The time between heel contact of the contralateral foot and just before toe off of the other foot.
Swing phase (s)	The period of time during which the foot has no contact with the ground.

**Table 2 sensors-22-02790-t002:** Intraclass Correlation Coefficients (ICCs) with 95% Confidence Intervals (95% CI) and systematic bias (SB) with 95% Limits of Agreement (95% LoA) for the spatiotemporal parameters measured by the OCS and the ITR during walking at an inclination of 0%.

Gait Parameter	Reliability	Minimum Sensors Used for Detection of the Contact Event				
(0% Slope)	Indices	1 LED	2 LEDs	3 LEDs	4 LEDs	5 LEDs
Step length (cm)	ICC (95% CI) SB (95% LoA)	0.998 (0.995, 0.999) 0.06 (−0.45, 0.57)	0.998 (0.995, 0.999) 0.06 (−0.45, 0.58)	0.998 (0.995, 0.999) 0.06 (−0.45, 0.58)	0.998 (0.995, 0.999) 0.07 (−0.45, 0.59)	0.998 (0.995, 0.999) 0.08 (−0.44, 0.59) ^b^
Stride length (cm)	ICC (95% CI) SB (95% LoA)	0.998 (0.996, 0.999) 0.04 (−0.87, 0.95)	0.998 (0.996, 0.999) 0.04 (−0.87, 0.95)	0.998 (0.996, 0.999) 0.05 (−0.85, 0.95)	0.998 (0.996, 0.999) 0.06 (−0.86, 0.98)	0.998 (0.996, 0.999) 0.06 (−0.85, 0.98)
Step time (s)	ICC (95% CI) SB (95% LoA)	0.998 (0.995, 0.999) 0.00 (0.00, 0.00)	0.998 (0.995, 0.999) 0.00 (0.00, 0.00)	0.998 (0.996, 0.999) 0.00 (0.00, 0.00)	0.998 (0.995, 0.999) 0.00 (0.00, 0.00)	0.998 (0.996, 0.999) 0.00 (0.00, 0.00)
Stride time (s)	ICC (95% CI) SB (95% LoA)	0.994 (0.987, 0.997) 0.00 (−0.01 0.01)	0.994 (0.987, 0.997) 0.00 (−0.01, 0.01)	0.994 (0.987, 0.997) 0.00 (−0.01, 0.01)	0.994 (0.987, 0.997) 0.00 (−0.01, 0.01)	0.994 (0.987, 0.997) 0.00 (−0.01, 0.01)
Cadence (steps/min)	ICC (95% CI) SB (95% LoA)	0.999 (0.998, 1.000) 0.08 (−0.47, 0.62)	0.999 (0.998, 1.000) 0.08 (−0.47, 0.62) ^c^	0.999 (0.998, 1.000) 0.07 (−0.48, 0.62) ^c,d^	0.999 (0.998, 1.000) 0.07 (−0.48, 0.62) ^c,d^	0.999 (0.998, 1.000) 0.07 (−0.48, 0.62) ^c,d^
Load response phase (s)	ICC (95% CI) SB (95% LoA)	0.871 (0.717, 0.941) 0.05 ^a^ (0.03, 0.07)	0.921 (0.826, 0.964) 0.04 ^a^ (0.02, 0.06) ^e^	0.911 (0.805, 0.960) 0.03 ^a^ (0.01, 0.04) ^e^	0.888 (0.755, 0.949) 0.01 ^a^ (−0.01, 0.03) ^e^	0.880 (0.737, 0.945) 0.00 (−0.02, 0.02) ^e^
Single support phase (s)	ICC (95% CI) SB (95% LoA)	0.878 (0.732, 0.944) −0.05 ^a^ (−0.07, −0.02)	0.901 (0.784, 0.955) −0.04 ^a^ (−0.06, −0.02) ^e^	0.888 (0.754, 0.949) −0.03 ^a^ (−0.05, −0.01) ^e^	0.868 (0.709, 0.940) −0.01 ^a^ (−0.03, 0.01) ^e^	0.865 (0.703, 0.938) 0.00 (−0.02, 0.02) ^e^
Pre-swing phase (s)	ICC (95% CI) SB (95% LoA)	0.882 (0.741, 0.946) 0.05 ^a^ (0.03, 0.07)	0.928 (0.841, 0.967) 0.04 ^a^ (0.02, 0.05) ^e^	0.918 (0.820, 0.963) 0.03 ^a^ (0.01, 0.04) ^e^	0.895 (0.770, 0.952) 0.01 ^a^ (−0.01, 0.03) ^e^	0.888 (0.754, 0.949) 0.00 (−0.02, 0.02) ^e^
Swing phase (s)	ICC (95% CI) SB (95% LoA)	0.885 (0.747, 0.947) −0.05 ^a^ (−0.07, −0.02)	0.908 (0.798, 0.958) −0.04 ^a^ (−0.06, −0.02) ^e^	0.895 (0.769, 0.952) −0.03 ^a^ (−0.05, −0.01) ^e^	0.870 (0.715, 0.941) −0.01 ^a^ (−0.03, 0.01) ^e^	0.868 (0.710, 0.940) 0.00 (−0.02, 0.02) ^e^

Color code designating excellent (green) and good (yellow) ICC reliability indices. ^a^ *p* < 0.001 for significant biases using 1, 2, 3, 4 and 5 LEDs. ^b^ *p* < 0.05 for 1 vs. 5 LEDs. ^c^ *p* ≤ 0.01 for 1 vs. 2; *p* ≤ 0.001 for 1 vs. 3, 4 and 5 LEDs. ^d^ *p* < 0.05 for 2 vs. 3 LEDs; *p* ≤ 0.001 for 2 vs. 4 and 5 LEDs. ^e^ *p* < 0.001 for 1 vs. 2, 3, 4 and 5 LEDs, 2 vs. 3, 4 and 5 LEDs, 3 vs. 4 and 5 LEDs and 4 vs. 5 LEDs.

**Table 3 sensors-22-02790-t003:** Intraclass Correlation Coefficients (ICCs) with 95% Confidence Intervals (95% CI) and systematic bias (SB) with 95% Limits of Agreement (95% LoA) for the spatiotemporal parameters measured by the OCS and the ITR during walking at an inclination of −10%.

Gait Parameter	Reliability	Minimum Sensors Used for Detection of the Contact Event				
(−10% Slope)	Indices	1 LED	2 LEDs	3 LEDs	4 LEDs	5 LEDs
Step length (cm)	ICC (95% CI) SB (95% LoA)	0.998 (0.995, 0.999) 0.04 (−0.53, 0.60)	0.997 (0.995, 0.999) 0.04 (−0.54, 0.62)	0.998 (0.995, 0.999) 0.04 (−0.52, 0.61)	0.997 (0.994, 0.999) 0.06 (−0.53, 0.65)	0.997 (0.994, 0.999) 0.07 (−0.51, 0.65) ^b^
Stride length (cm)	ICC (95% CI) SB (95% LoA)	0.999 (0.997, 0.999) 0.21^a^ (−0.62, 1.04)	0.999 (0.997, 0.999) 0.23 ^a^ (−0.61, 1.07)	0.999 (0.997, 0.999) 0.25 ^a^ (−0.62, 1.12)	0.999 (0.997, 0.999) 0.26 ^a^ (−0.62, 1.15)	0.998 (0.997, 0.999) 0.26 ^a^ (−0.63, 1.16)
Step time (s)	ICC (95% CI) SB (95% LoA)	0.996 (0.992, 0.998) 0.00 (0.00, 0.00)	0.997 (0.993, 0.998) 0.00 (0.00, 0.00)	0.997 (0.992, 0.998) 0.00 (0.00, 0.00)	0.996 (0.992, 0.998) 0.00 (0.00, 0.00)	0.997 (0.993, 0.998) 0.00 (0.00, 0.00)
Stride time (s)	ICC (95% CI) SB (95% LoA)	0.999 (0.997, 0.999) 0.00 (−0.01, 0.01)	0.999 (0.997, 0.999) 0.00 (0.01, 0.01)	0.999 (0.997, 0.999) 0.00 (−0.01, 0.01)	0.999 (0.997, 0.999) 0.00 (−0.01, 0.01)	0.999 (0.997, 0.999) 0.00 (−0.01, 0.01)
Cadence (steps/min)	ICC (95% CI) SB (95% LoA)	0.999 (0.998, 1.000) 0.08 (−0.54, 0.70)	0.999 (0.998, 1.000) 0.08 (−0.54, 0.69) ^c^	0.999 (0.998, 1.000) 0.07 (−0.55, 0.69) ^c,d^	0.999 (0.998, 1.000) 0.07 (−0.55, 0.69) ^c,d,e^	0.999 (0.998, 1.000) 0.07 (−0.55, 0.68) ^c,d,e^
Load response phase (s)	ICC (95% CI) SB (95% LoA)	0.921 (0.826, 0.964) 0.04 ^a^ (0.03, 0.06)	0.938 (0.865, 0.972) 0.04 ^a^ (0.02, 0.05) ^f^	0.920 (0.824, 0.963) 0.03 ^a^ (0.01, 0.04) ^f^	0.903 (0.786, 0.956) 0.01 ^a^ (0.00, 0.03) ^f^	0.899 (0.779, 0.954) 0.00 (−0.02, 0.02) ^f^
Single support phase (s)	ICC (95% CI) SB (95% LoA)	0.952 (0.895, 0.978) −0.04 ^a^ (−0.06, −0.03)	0.964 (0.920, 0.983) −0.04 ^a^ (−0.05, −0.02) ^f^	0.952 (0.896, 0.978) −0.03 ^a^ (−0.04, −0.01) ^f^	0.944 (0.878, 0.975) −0.01 ^a^ (−0.03, 0.00) ^f^	0.940 (0.869, 0.973) 0.00 (−0.02, 0.01) ^f^
Pre-swing phase (s)	ICC (95% CI) SB (95% LoA)	0.924 (0.834, 0.965) 0.04 ^a^ (0.03, 0.06)	0.941 (0.870, 0.973) 0.04 ^a^ (0.02, 0.05) ^f^	0.923 (0.832, 0.965) 0.03 ^a^ (0.01, 0.04) ^f^	0.905 (0.792, 0.957) 0.01 ^a^ (0.00, 0.03) ^f^	0.899 (0.779, 0.954) 0.00 (−0.02, 0.02) ^f^
Swing phase (s)	ICC (95% CI) SB (95% LoA)	0.952 (0.894, 0.978) −0.04 ^a^ (−0.06, −0.03)	0.963 (0.919, 0.983) −0.04 ^a^ (−0.05, −0.02) ^f^	0.952 (0.895, 0.978) −0.03 ^a^ (−0.04, −0.01) ^f^	0.944 (0.877, 0.975) −0.01 ^a^ (−0.03, 0.00) ^f^	0.940 (0.868, 0.973) 0.00 (−0.02, 0.01) ^f^

Color code designating excellent (green) and good (yellow) ICC reliability indices. ^a^ *p* < 0.05 for 1 LED, *p* < 0.01 for 2, 3, 4, and 5 LEDs; *p* < 0.001 for 1, 2, 3 and 4 LEDs in gait phases. ^b^ *p* < 0.05, for 1 vs. 5 LEDs; *p* < 0.01, for 2 vs. 5 LEDs; *p* < 0.001, for 3 vs. 5 LEDs. ^c^ *p* < 0.01, for 1 vs. 2 LEDs; *p* < 0.001, for 1 vs. 3, 4 and 5 LEDs. ^d^ *p* ≤ 0.001, for 2 vs. 3, 4 and 5 LEDs. ^e^ *p* < 0.05, for 3 vs. 4 LEDs; *p* < 0.01, for 3 vs. 5 LEDs. ^f^ *p* < 0.001 for 1 vs. 2, 3, 4 and 5 LEDs, 2 vs. 3, 4 and 5 LEDs, 3 vs. 4 and 5 LEDs and 4 vs. 5 LEDs.

**Table 4 sensors-22-02790-t004:** Intraclass Correlation Coefficients (ICCs) with 95% Confidence Intervals (95% CI) and systematic bias (SB) with 95% Limits of Agreement (95% LoA) for the spatiotemporal parameters measured by the OCS and the ITR during walking at an inclination of −20%.

Gait Parameter	Reliability	Minimum Sensors Used for Detection of the Contact Event				
(−20% Slope)	Indices	1 LED	2 LEDs	3 LEDs	4 LEDs	5 LEDs
Step length (cm)	ICC (95% CI) SB (95% LoA)	0.997 (0.994, 0.999) 0.06 (−0.50, 0.63)	0.997 (0.993, 0.999) 0.06 (−0.52, 0.63)	0.997 (0.994, 0.999) 0.06 (−0.50, 0.62)	0.997 (0.993, 0.999) 0.06 (−0.51, 0.64)	0.997 (0.994, 0.999) 0.05 (−0.52, 0.62)
Stride length (cm)	ICC (95% CI) SB (95% LoA)	0.997 (0.993, 0.999) 0.15 (−1.03, 1.33)	0.997 (0.993, 0.999) 0.15 (−1.03, 1.33)	0.997 (0.993, 0.999) 0.15 (−1.03, 1.33)	0.997 (0.993, 0.999) 0.15 (−1.02, 1.33)	0.997 (0.993, 0.999) 0.15 (−1.03, 1.33)
Step time (s)	ICC (95% CI) SB (95% LoA)	0.994 (0.987, 0.997) 0.00 (−0.01, 0.01)	0.997 (0.994, 0.999) 0.00 (−0.01, 0.00)	0.998 (0.995, 0.999) 0.00 (−0.01, 0.00)	0.998 (0.996, 0.999) 0.00 (−0.01, 0.00)	0.998 (0.996, 0.999) 0.00 (−0.01, 0.00)
Stride time (s)	ICC (95% CI) SB (95% LoA)	0.979 (0.954, 0.990) 0.00 (−0.04, 0.03)	0.979 (0.955, 0.991) 0.00 (−0.04, 0.03)	0.979 (0.955, 0.991) 0.00 (−0.03, 0.03)	0.980 (0.955, 0.991) 0.00 (−0.03, 0.03)	0.980 (0.955, 0.991) 0.00 (−0.03, 0.03)
Cadence (steps/min)	ICC (95% CI) SB (95% LoA)	0.984 (0.966, 0.993) −0.47 (−3.82, 2.88)	0.996 (0.991, 0.998) −0.27 (−2.08, 1.54)	0.997 (0.994, 0.999) −0.23 (−1.69, 1.23)	0.999 (0.997, 0.999) −0.14 (−1.20, 0.92)	0.999 (0.997, 0.999) −0.13 (−1.13, 0.88)
Load response phase (s)	ICC (95% CI) SB (95% LoA)	0.896 (0.771, 0.952) 0.05 ^a^ (0.02, 0.07)	0.908 (0.798, 0.958) 0.04 ^a^ (0.02, 0.06) ^b^	0.916 (0.816, 0.962) 0.03 ^a^ (0.01, 0.05) ^b^	0.893 (0.764, 0.951) 0.02 ^a^ (0.00, 0.04) ^b^	0.865 (0.704, 0.938) 0.01 ^a^ (−0.01, 0.03) ^b^
Single support phase (s)	ICC (95% CI) SB (95% LoA)	0.983 (0.863, 0.972) −0.05 ^a^ (−0.07, −0.03)	0.946 (0.881, 0.975) −0.04 ^a^ (−0.06, −0.02) ^b^	0.950 (0.891, 0.977) −0.03 ^a^ (−0.05, −0.01) ^b^	0.950 (0.890, 0.977) −0.02 ^a^ (−0.04, 0.00) ^b^	0.946 (0.882, 0.975) −0.01 ^a^ (−0.03, 0.01) ^b^
Pre-swing phase (s)	ICC (95% CI) SB (95% LoA)	0.893 (0.765, 0.951) 0.05 ^a^ (0.02, 0.07)	0.907 (0.797, 0.958) 0.04 ^a^ (0.02, 0.06) ^b^	0.915 (0.813, 0.961) 0.03 ^a^ (0.01, 0.05) ^b^	0.892 (0.763, 0.951) 0.02 ^a^ (0.00, 0.04) ^b^	0.864 (0.701, 0.938) 0.01 ^a^ (−0.01, 0.03) ^b^
Swing phase (s)	ICC (95% CI) SB (95% LoA)	0.938 (0.865, 0.972) −0.05 ^a^ (−0.07, −0.03)	0.946 (0.882, 0.976) −0.04 ^a^ (−0.06, −0.02) ^b^	0.951 (0.893, 0.978) −0.03 ^a^ (−0.05, −0.01) ^b^	0.950 (0.891, 0.977) −0.02 ^a^ (−0.04, 0.00) ^b^	0.947 (0.883, 0.976) −0.01 ^a^ (−0.03, 0.01) ^b^

Color code designating excellent (green) and good (yellow) ICC reliability indices. ^a^ *p* < 0.01 for 5 LEDs (for Load response and Pre-swing phases; *p* < 0.001 for 1, 2, 3, 4, and 5 LEDs (for Single support and Swing phases). ^b^ *p* < 0.001 for 1 vs. 2, 3, 4 and 5 LEDs, 2 vs. 3, 4 and 5 LEDs, 3 vs. 4 and 5 LEDs and 4 vs. 5 LEDs.

**Table 5 sensors-22-02790-t005:** Intraclass Correlation Coefficients (ICCs) with 95% Confidence Intervals (95% CI) and systematic bias (SB) with 95% Limits of Agreement (95% LoA) for the spatiotemporal parameters measured by the OCS and the ITR during walking at an inclination of +10%.

Gait Parameter	Reliability	Minimum Sensors Used for Detection of the Contact Event				
(+10% Slope)	Indices	1 LED	2 LEDs	3 LEDs	4 LEDs	5 LEDs
Step length (cm)	ICC (95% CI) SB (95% LoA)	0.998 (0.996, 0.999) −0.21 ^a^ (−0.83, 0.41)	0.998 (0.996, 0.999) −0.22 ^a^ (−0.85, 0.42)	0.998 (0.995, 0.999) −0.22 ^a^ (−0.87, 0.43)	0.998 (0.995, 0.999) −0.23 ^a^ (−0.89, 0.44)	0.997 (0.994, 0.999) −0.26 ^a^ (−0.97, 0.46)
Stride length (cm)	ICC (95% CI) SB (95% LoA)	0.999 (0.998, 1.000) −0.41 ^b^ (−1.29, 0.46)	0.999 (0.998, 1.000) −0.41 ^b^ (−1.33, 0.50)	0.999 (0.998, 0.999) −0.43 ^b^ (−1.38, 0.51)	0.999 (0.997, 0.999) −0.45 ^b^ (−1.45, 0.55)	0.999 (0.997, 0.999) −0.46 ^b^ (−1.51, 0.58)
Step time (s)	ICC (95% CI) SB (95% LoA)	0.987 (0.971, 0.994) 0.00 (−0.01, 0.02)	0.987 (0.971, 0.994) 0.00 (−0.01, 0.02)	0.987 (0.971, 0.994) 0.00 (−0.01, 0.02)	0.987 (0.971, 0.994) 0.00 (−0.01, 0.02)	0.987 (0.971, 0.994) 0.00 (−0.01, 0.02)
Stride time (s)	ICC (95% CI) SB (95% LoA)	0.989 (0.977, 0.995) 0.00 (−0.03, 0.03)	0.989 (0.977, 0.995) 0.00 (−0.03, 0.03)	0.989 (0.977, 0.995) 0.00 (−0.03, 0.03)	0.989 (0.977, 0.995) 0.00 (−0.03, 0.03)	0.989 (0.977, 0.995) 0.00 (−0.03, 0.03)
Cadence (steps/min)	ICC (95% CI) SB (95% LoA)	0.999 (0.999, 1.000) 0.08 (−0.49, 0.64)	0.999 (0.999, 1.000) 0.07 (−0.49, 0.64) ^d^	0.999 (0.999, 1.000) 0.07 (−0.49, 0.63)	0.999 (0.999, 1.000) 0.07 (−0.49, 0.63) ^e^	0.999 (0.999, 1.000) 0.07 (−0.49, 0.63)
Load response phase (s)	ICC (95% CI) SB (95% LoA)	0.884 (0.747, 0.947) 0.08 ^c^ (0.06, 0.11)	0.890 (0.759, 0.950) 0.07 ^c^ (0.05, 0.10) ^f^	0.891 (0.760, 0.950) 0.05 ^c^ (0.03, 0.08) ^f^	0.871 (0.717, 0.941) 0.03 ^c^ (0.01, 0.06) ^f^	0.867 (0.708, 0.939) 0.01 ^c^ (−0.02, 0.04) ^f^
Single support phase (s)	ICC (95% CI) SB (95% LoA)	0.918 (0.819, 0.962) −0.08 ^c^ (−0.11, −0.05)	0.932 (0.850, 0.969) −0.07 ^c^ (−0.10, −0.04) ^f^	0.934 (0.856, 0.970) −0.05 ^c^ (−0.08, −0.03) ^f^	0.931 (0.848, 0.969) −0.03 ^c^ (−0.06, 0.00) ^f^	0.932 (0.851, 0.969) −0.01 ^c^ (−0.04, 0.02) ^f^
Pre-swing phase (s)	ICC (95% CI) SB (95% LoA)	0.882 (0.742, 0.946) 0.08 ^c^ (0.06, 0.11)	0.889 (0.756, 0.949) 0.07 ^c^ (0.05, 0.10) ^f^	0.891 (0.761, 0.950) 0.05 ^c^ (0.03, 0.08) ^f^	0.870 (0.715, 0.941) 0.03 ^c^ (0.01, 0.06) ^f^	0.866 (0.706, 0.939) 0.01 ^c^ (−0.02, 0.04) ^f^
Swing phase (s)	ICC (95% CI) SB (95% LoA)	0.920 (0.825, 0.964) −0.08 ^c^ (−0.11, −0.05)	0.934 (0.855, 0.970) −0.07 ^c^ (−0.10, −0.04) ^f^	0.935 (0.858, 0.971) −0.05 ^c^ (−0.08, −0.03) ^f^	0.930 (0.847, 0.968) −0.03 ^c^ (−0.06, 0.00) ^f^	0.931 (0.849, 0.969) −0.01 ^c^ (−0.04, 0.02) ^f^

Color code designating excellent (green) and good (yellow) ICC reliability indices. ^a^ *p* < 0.01 significant biases for 1, 3 and 4 LEDs, *p* ≤ 0.001 for 2 and 5 LEDs. ^b^ *p* < 0.001 significant biases for 1, 2, 3, 4, and 5 LEDs. ^c^ *p* < 0.01 for 5 LEDs (for Single support and Swing phases); *p* < 0.001 significant biases for 1, 2, 3, 4 and 5 LEDs (for Load response and Pre-swing phases). ^d^ *p* < 0.05 for 1 vs. 2 LEDs. ^e^ *p* < 0.01 for 1 vs. 4 LEDs and *p* < 0.05 for 3 vs. 4 LEDs. ^f^ *p* < 0.001 for 1 vs. 2, 3, 4 and 5 LEDs, 2 vs. 3, 4 and 5 LEDs, 3 vs. 4 and 5 LEDs and 4 vs. 5 LEDs.

**Table 6 sensors-22-02790-t006:** Intraclass Correlation Coefficients (ICCs) with 95% Confidence Intervals (95% CI) and systematic bias (SB) with 95% Limits of Agreement (95% LoA) for the spatiotemporal parameters measured by the OCS and the ITR during walking at an inclination of +20%.

Gait Parameter	Reliability	Minimum Sensors Used for Detection of the Contact Event				
(+20% Slope)	Indices	1 LED	2 LEDs	3 LEDs	4 LEDs	5 LEDs
Step length (cm)	ICC (95% CI) SB (95% LoA)	0.998 (0.996, 0.999) 0.24 ^a^ (−0.46, 0.93)	0.998 (0.996, 0.999) 0.25 ^a^ (−0.47, 0.96)	0.998 (0.996, 0.999) 0.24 ^a^ (−0.47, 0.95)	0.998 (0.996, 0.999) 0.24 ^a^ (−0.47, 0.96)	0.998 (0.996, 0.999) 0.23 ^a^ (−0.47, 0.93)
Stride length (cm)	ICC (95% CI) SB (95% LoA)	0.998 (0.996, 0.999) 0.58 ^b^ (−0.83, 2.00)	0.999 (0.996, 0.999) 0.59 ^b^ (−0.86, 2.04)	0.998 (0.996, 0.999) 0.58 ^b^ (−0.84, 2.00)	0.998 (0.996, 0.999) 0.59 ^b^ (−0.84, 2.03)	0.998 (0.996, 0.999) 0.57 ^b^ (−0.78, 1.93)
Step time (s)	ICC (95% CI) SB (95% LoA)	0.999 (0.999, 1.000) 0.00 (−0.01, 0.00)	0.999 (0.999, 1.000) 0.00 (−0.01, 0.01)	0.999 (0.999, 1.000) 0.00 (−0.01, 0.01)	0.999 (0.998, 1.000) 0.00 (−0.01, 0.01) ^e^	0.999 (0.998, 1.000) 0.00 (−0.01, 0.01)
Stride time (s)	ICC (95% CI) SB (95% LoA)	0.999 (0.998, 1.000) 0.00 (−0.01, 0.02)	0.999 (0.998, 1.000) 0.00 (−0.01, 0.02)	0.999 (0.998, 1.000) 0.00 ^c^ (−0.01, 0.02)	0.999 (0.998, 1.000) 0.00 ^c^ (−0.01, 0.02)	0.999 (0.998, 1.000) 0.00 ^c^ (−0.01, 0.02)
Cadence (steps/min)	ICC (95% CI) SB (95% LoA)	0.996 (0.991, 0.998) −0.29 (−2.19, 1.60)	0.997 (0.994, 0.999) −0.21 (−1.82, 1.41)	0.998 (0.996, 0.999) −0.11 (−1.47, 1.25)	0.998 (0.996, 0.999) −0.10 (−1.43, 1.24)	0.999 (0.997, 0.999) −0.07 (−1.23, 1.09)
Load response phase (s)	ICC (95% CI) SB (95% LoA)	0.968 (0.929, 0.985) 0.09 ^d^ (0.06, 0.11)	0.969 (0.933, 0.986) 0.08 ^d^ (0.05, 0.10) ^f^	0.961 (0.914, 0.982) 0.06 ^d^ (0.04, 0.09) ^f^	0.952 (0.894, 0.978) 0.04 ^d^ (0.02, 0.07) ^f^	0.918 (0.819, 0.962) 0.03 ^d^ (0.00, 0.06) ^f^
Single support phase (s)	ICC (95% CI) SB (95% LoA)	0.950 (0.890, 0.977) −0.08 ^d^ (−0.12, −0.05)	0.956 (0.904, 0.980) −0.07 ^d^ (−0.10, −0.04) ^f^	0.960 (0.911, 0.982) −0.06 ^d^ (−0.09, −0.03) ^f^	0.967 (0.927, 0.985) −0.04 ^d^ (−0.07, −0.01) ^f^	0.962 (0.917, 0.983) −0.03 ^d^ (−0.06, 0.00) ^f^
Pre-swing phase (s)	ICC (95% CI) SB (95% LoA)	0.966 (0.925, 0.985) 0.09 ^d^ (0.06, 0.11)	0.969 (0.931, 0.986) 0.08 ^d^ (0.05, 0.10) ^f^	0.960 (0.913, 0.982) 0.06 ^d^ (0.04, 0.09) ^f^	0.949 (0.888, 0.977) 0.04 ^d^ (0.02, 0.07) ^f^	0.915 (0.813, 0.961) 0.03 ^d^ (0.00, 0.06) ^f^
Swing phase (s)	ICC (95% CI) SB (95% LoA)	0.952 (0.895, 0.978) −0.08 ^d^ (−0.11, −0.05)	0.959 (0.911, 0.982) −0.07 ^d^ (−0.10, −0.05) ^f^	0.962 (0.916, 0.983) −0.06 ^d^ (−0.09, −0.03) ^f^	0.967 (0.928, 0.985) −0.04 ^d^ (−0.07, −0.01) ^f^	0.961 (0.915, 0.982) −0.03 ^d^ (−0.06, 0.00) ^f^

Color code designating excellent (green) ICC reliability indices. ^a^ *p* < 0.01 significant biases for 1, 3, 4 and 5 LEDs; *p* ≤ 0.001 for 2 LEDs. ^b^
*p* < 0.001 significant biases for 1, 2, 3 4 and 5 LEDs. ^c^
*p* < 0.001 significant biases for 3, 4 and 5 LEDs. ^d^
*p* < 0.001 significant biases for 1, 2, 3, 4 and 5 LEDs. ^e^
*p* < 0.05 for 1 vs. 4 LEDs. ^f^
*p* < 0.001 for 1 vs. 2, 3, 4 and 5 LEDs, 2 vs. 3, 4 and 5 LEDs, 3 vs. 4 and 5 LEDs and 4 vs. 5 LEDs.

## Data Availability

The data presented in this study are available upon request from the corresponding author.

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
