# Peer review of "Differences between Systems Using Optical and Capacitive Sensors in Treadmill-Based Spatiotemporal Analysis of Level and Sloping Gait"

_sensors, 2022, doi:10.3390/s22072790_

Round 1

Reviewer 1 Report

In this paper, the authors investigate the agreement between systems equipped with optical and capacitive sensors in the analysis of the treadmill-based level and sloping gait. The spatiotemporal parameters of gait were measured in 30 healthy college-level students during barefoot walking on 0% (level), -10% and -20% (downhill), and +10% and +20% (uphill) slopes at hiking-related speeds using an opt electric cell system and an instrumented treadmill. Which can be used interchangeably in the treadmill-based spatiotemporal analysis of the level and sloping gait. This article is clear, concise, and suitable for the scope of the journal. One small suggestion: Suggest the authors supply a schematic for the experiments, making it more readable and more attractive. Also, supply some graphs about the experiment setup. 

Author Response

Dear Editor/Dear Reviewers

We would like to thank you for considering the article titled “Differences between Systems Using Optical and Capacitive Sensors in Treadmill-based Spatiotemporal Analysis of Level and Sloping Gait” for publication in the “Sensor” journal, and for the time you spent reviewing it providing us with helpful and constructive feedback on optimizing our article. All comments were noted and appropriate modifications were made and highlighted in gray in the current version of the manuscript in order to comply with the reviewers’ suggestions.

Reviewer 1

Reviewer’s suggestion: …supply a schematic for the experiments, making it more readable and more attractive and some graphs about the experiment setup.

Authors’ respond: A schematic for the experiment (page 5), and graphs about the experiment setup (page 3) were provided as suggested by the reviewer.

Dimitris Mandalidis, Associate professor

Sports Physical Therapy Laboratory

Department of Physical Education and Sports Science

School of Physical Education and Sports Science

National and Kapodistrian University of Athens

Reviewer 2 Report

Mandalidis and Kafetzakis compared the performance of a low-cost optical gait measurement device that can be used as an add-on to a treadmill (Optogait) with a capacitance-based pressure platform. This study suggests that these two systems have comparable performance and can be used interchangeably. The authors also explored different parameters (e.g., slope and number of LEDs) and assessed their effect on the similarity between the parameters measured by two different systems. 

In general, this study is well-designed and its results provide sufficient evidence that can be used by other researchers and clinicians for gait assessment.

The only shortcoming of the paper is the results section:

1- The data can be illustrated in more informative forms. For example, the systematic bias can be shown as point-of-bar plots to better reflect the trends described by the text. If it is possible the different levels of ICC (i.e., poor, good, and excellence) can be color-coded (e.g, red, yellow, green) to convey the results much faster.

2 - The data presented in the figures were not explained properly in the figure legend. The meaning of each point and color is not clear. More importantly, none of the figures has been mentioned in the results text at all. The authors should add text corresponding to the figures in the results section.    

Author Response

Dear Editor/Dear Reviewers

We would like to thank you for considering the article titled “Differences between Systems Using Optical and Capacitive Sensors in Treadmill-based Spatiotemporal Analysis of Level and Sloping Gait” for publication in the “Sensor” journal, and for the time you spent reviewing it providing us with helpful and constructive feedback on optimizing our article. All comments were noted and appropriate modifications were made and highlighted in gray in the current version of the manuscript in order to comply with the reviewers’ suggestions.

Reviewer 2

Reviewer’s suggestion: The data can be illustrated in more informative forms. For example, the systematic bias can be shown as point-of-bar plots to better reflect the trends described by the text. If it is possible the different levels of ICC (i.e., poor, good, and excellence) can be color-coded (e.g, red, yellow, green) to convey the results much faster.

Authors’ respond: Due to the large volume of data, it was decided to list all the results (ICC and 95%LoA) in tables. Systematic bias and 95% LoA for step and gait length as well as cadence showed small changes by increasing the sensors for gait analysis with OCS while the step and stride time was not affected at all. Only the duration of the gait phases presented trends (changes with the number of LEDs used) that could be more informative if visualized and therefore only the systematic biases and 95% LoA of these parameters were presented with graphs. For this reason we used the Bland and Altman’s plot (differences between measurements, i.e. systematic bias vs. average between measurements), which depicts both the systematic biases and the 95% LoA resulting from measurements between measuring devices. In this context, we would like to retain the graphs that we submitted with the first version of the manuscript, modified based on the reviewer’s suggestions (see the authors' response in the next section). Furthermore, a color code was used to identify the different levels of ICC as proposed by the reviewer.

Reviewer’s suggestion: The data presented in the figures were not explained properly in the figure legend. The meaning of each point and color is not clear. More importantly, none of the figures has been mentioned in the results text at all. The authors should add text corresponding to the figures in the results section.    

Authors’ respond: The figures present the Bland and Altman plots (differences between measurements, i.e. systematic bias vs. average between measurements), which depicts both the systematic biases and the 95% LoA. Their legends were modified in order to be more explanatory regarding the content of the figures. Additionally, each point and color in the plot was identified. Finally, all the figures were mentioned in the text as indicated by the reviewer.

Dimitris Mandalidis, Associate professor

Sports Physical Therapy Laboratory

Department of Physical Education and Sports Science

School of Physical Education and Sports Science

National and Kapodistrian University of Athens
